# Invertebrate Immunity, Natural Transplantation Immunity, Somatic and Germ Cell Parasitism, and Transposon Defense

**DOI:** 10.3390/ijms25021072

**Published:** 2024-01-16

**Authors:** Malgorzata Kloc, Marta Halasa, Jacek Z. Kubiak, Rafik M. Ghobrial

**Affiliations:** 1Houston Methodist Research Institute, Transplant Immunology, Houston, TX 77030, USA; mhalasa@houstonmethodist.org (M.H.); rmghobrial@houstonmethodist.org (R.M.G.); 2Department of Surgery, Houston Methodist Hospital, Houston, TX 77030, USA; 3Department of Genetics, MD Anderson Cancer Center, University of Texas, Houston, TX 77030, USA; 4Laboratory of Molecular Oncology and Innovative Therapies, Military Institute of Medicine-National Research Institute (WIM-PIB), Szaserow 128, 04-141 Warsaw, Poland; jacek.kubiak@univ-rennes.fr; 5Dynamics and Mechanics of Epithelia Group, Faculty of Medicine, Institute of Genetics and Development of Rennes, University of Rennes, CNRS, UMR 6290, 35043 Rennes, France

**Keywords:** innate immunity, innate memory, hemocyte, invertebrate, transposons, epigenetics, transgenerational inheritance

## Abstract

While the vertebrate immune system consists of innate and adaptive branches, invertebrates only have innate immunity. This feature makes them an ideal model system for studying the cellular and molecular mechanisms of innate immunity *sensu stricto* without reciprocal interferences from adaptive immunity. Although invertebrate immunity is evolutionarily older and a precursor of vertebrate immunity, it is far from simple. Despite lacking lymphocytes and functional immunoglobulin, the invertebrate immune system has many sophisticated mechanisms and features, such as long-term immune memory, which, for decades, have been exclusively attributed to adaptive immunity. In this review, we describe the cellular and molecular aspects of invertebrate immunity, including the epigenetic foundation of innate memory, the transgenerational inheritance of immunity, genetic immunity against invading transposons, the mechanisms of self-recognition, natural transplantation, and germ/somatic cell parasitism.

## 1. Introduction

All organisms, even unicellular ones, must have the mechanisms to distinguish the self from the non-self and defend against pathogens. Progressive improvements of such mechanisms over millions of years of evolution culminated in a marvel—the jawed vertebrates’ immune system with two mechanistically distinct but interacting branches: innate and adaptive immunity. While innate immunity is the fast and generic first line of defense, adaptive immunity is specific, anticipatory, long-lasting, and able to memorize previous encounters with antigens. This traditional view of the strict division of immune defense tasks does not hold ground in the view of recent advancements in molecular and cellular studies [1]. In the traditional view, innate immunity relies on motile, ameba-like cells (amebocytes, hemocytes, coelomocytes, monocytes, and macrophages), which, for many decades, were assumed to perform ancestral, straightforward functions of moving toward the target (pathogen or any foreign object), engulfing it, and digesting (destroying) it. However, many studies, especially those on invertebrates (worms, insects, crustaceans, chelicerates, mollusks, echinoderms, and tunicates) whose immune system consists only of the innate branch, showed that such a simplistic view of innate immune cells is inaccurate and that these cells can also memorize previous encounters with pathogens and develop lasting immune memory—the features of adaptive immunity [2,3,4]. Such innate immune memory, also called “trained immunity”, supplements adaptive immune memory and regulates adaptive responses (in organisms that have both immune branches} and involves cell metabolism modifications and epigenetic reprogramming. Another aspect of innate immunity is the humoral response, traditionally attributed to antibody-producing B cells, that acts through the functional ancestors of antibodies, such as pentraxins, collectins, and ficolins, which activate the complement, opsonize, agglutinate, and kill microorganisms [5,6,7,8]. On the other hand, adaptive immune cells (B and lymphocytes) can recognize and phagocytose targets using pattern recognition, thus performing functions originally attributed to innate immune cells [1,9,10].

Despite immunologic commonalities between different invertebrate groups, genomic studies showed that even closely related invertebrate species employ very different approaches in dealing with these universal challenges, which, in turn, suggests that invertebrate immune systems have evolved many times, independently from a common ancient precursor [1,4,11,12,13].

The following paragraphs describe some of the complexities of the invertebrate immune system and the role of natural transplantation immunity, somatic and germ cell parasitism, and transposon defense mechanisms.

## 2. Invertebrate Immunity

Although invertebrates are tremendously diversified in morphology, body plan, lifestyles, lifespan, and evolutionary history, all must cope with similar infectious agents (viruses, bacteria, protists, and fungi) and recognize genetically identical and allogeneic individuals. In invertebrates and vertebrates, the recognition of the invading pathogens is based on the same principle of all pathogenic organisms having certain molecular components that are different from those in the host. All these pathogen-specific components, which are usually located on the pathogen’s surface, are collectively called pathogen-related molecular patterns (PAMPs). There are many categories of PAMPs, such as various polysaccharides, lipids, peptides, proteins, and nucleic acids. PAMPs are recognized by the pattern recognition receptors (PRRs) expressed by innate immune cells (Figure 1). Different invertebrate species have different PRRs. For example, the PRRs typical for Lepidoptera are C-type lectin (CTL), peptidoglycan recognition protein (PGRP), β-1,3-glucan recognition protein (βGRP), and Gram-negative binding protein (GNBP) [11,14]. Other compounds also recognized by PRRs are damage-associated molecular patterns (DAMPs)—the molecules released from stressed, damaged, or dying cells, i.e., they represent an endogenous danger signal. Protein DAMPs include heat-shock proteins and molecules derived from the extracellular matrix of injured tissues, and non-protein DAMPs include nuclear and mitochondrial DNA, RNAs, ATP, components of ER, and cell membranes. Although DAMPs and PAMPs activate similar inflammatory responses, those caused by DAMPs are called the aseptic (infection-unrelated) type of inflammation. One of the insect-specific DAMPs is the dorsal switch protein 1 (DSP1)—an ortholog of mammalian high-mobility group box 1 protein (HMGB, amphoterin) released during autophagy [14]. Other groups of receptors recognizing both PAMPs and DAMPs and that are common between vertebrates and invertebrates are Toll-like receptors (TLRs) and scavenger receptors (SRs). Some TLRs are known to have a very high affinity to specific PAMPs. For example, TLR1 and TLR2 have a high affinity for bacterial lipoproteins and peptidoglycans, TLR3 has a high affinity for viral double-stranded RNA, TLR5 has a high affinity for bacterial flagellin, TLR9 has a high affinity for the unmethylated CpG of viral and bacterial DNA, and TLr11 and 12 have high affinity for bacterial profilin [15]. TLRs occur in all invertebrate groups, and because they are also present in the simplest invertebrates, such as cnidarians and sponges, they are evolutionarily the most ancient receptors [11]. For example, in shrimps, the *PmToll* receptor in *Panaeus monodon*, *lToll* in *Litopenaeus vannamei*, and *MjToll* in *Marsupenaeus japonicus* have been cloned and described functionally [16,17,18,19]. The SRs are diversified structurally, which allows them to bind a wide range of ligands and cooperate with other PRRs, such as TLRs, in activating phagocytosis and inflammation pathways [11,20,21,22,23]. Recently, various PRRs, including TRLs, have been identified in the transcriptome of the dung beetle *Copris tripartitus* [24] and giant African snail *Achatina immaculata* [25]. In *Drosophila* embryo, macrophages and hemocytes express scavenger receptor SR-CI (dSR-CI) [26], and shrimp *M. japonicus* has the B class type III scavenger receptor, SRB2, which induces an immune deficiency (IMD) pathway (see below) [27]. IMD and antimicrobial (AMP) pathway components were also described recently in the transcriptome of tick *Amblyomma americanum* [28].

The ability to make antimicrobial compounds (AMPs) is one of the hallmark functions of invertebrate innate immune cells. AMPs are made by all invertebrate and vertebrate species. AMPs are usually hydrophobic and cationic and effective against a wide range of microorganisms. A detailed description of various invertebrate and vertebrate AMPs and their promise for therapeutic use is given in [29,30]. Pathways regulating AMP production are especially well described in insects such as *Drosophila*, *Tenebrio*, and *Triatoma*. Two main pathways mediated by the nuclear transcription factor-kappa B (NF-κB) are as follows: Toll pathway against Gram-positive bacteria and fungi and the immune deficiency (IMD, named after the mutation in *Drosophila* causing severe immune deficiency) pathway against Gram-negative bacteria [31,32,33,34,35,36]. In the Toll pathway, the TLRs, upon recognizing PAMPs, recruit specific adaptor molecules and activate transcription factors NF-κB and, ultimately, the expression of proinflammatory genes and AMPs. In the IMD pathway, the recognition of PAMPs by PRRs, such as PGRP-LC and PGRP-LE, leads to the formation of PGRP homo- and heterodimers and the recruitment of the IMD protein, which in insects shares similarities with the receptor-interacting protein (RIP) of the mammalian tumor necrosis factor receptor (TNFR), followed by the activation of intracellular signaling and the phosphorylation and cleaving of NF-κB transcription factor Relish into N-terminal (Rel-68) and C terminal (Rel-49) fragments. Subsequently, Rel-68 translocates into the nucleus and activates the expression of AMPs [35,37].

Another form of antimicrobial defense is the programmed cell death of immune cells (ETosis), in which, in response to a pathogen, nuclear chromatin is decondensed and dispelled from immune cells. The strands of released DNA create web-like structures (extracellular traps, ETs) decorated with antimicrobial compounds, which capture and kill pathogens. ETosis occurs in the neutrophiles (NETosis) and macrophages (METosis) of vertebrates and the hemocytes of invertebrates [38,39,40].

Additionally, antimicrobial defense is supported by the proteasomal and autophagy pathways. Proteasomes are giant cytoplasmic and nuclear multi-subunit proteases that digest proteins tagged for degradation via polyubiquitination. The proteasome ubiquitination pathway occurs in all animals and plants and, by targeting foreign proteins for proteolysis, defends against microorganisms and bacterial protein toxins [41,42]. Similarly, in the defense against invading pathogens, macroautophagy and autophagy eliminate intracellular components and pathogens through a lysosome-dependent degradation pathway and activate a series of immune responses through pattern recognition receptors (PRRs) [43,44].

## 3. Invertebrate Immune Cells

Except for the simplest invertebrates, such as cnidarians, most invertebrates contain immune cells. However, despite lacking specialized, mobile immune cells, cnidarians, such as *Hydra,* have a robust immune defense that is sometimes called “the mucosal immunity” because of its similarity to the immune responses of epithelia in the gastrointestinal and upper respiratory tracts’ mucosa in vertebrates [45,46,47]. *Hydra* has tissues but does not have organs. The *Hydra* body is a simple tube built from two epithelial (ectoderm and endoderm) layers with interspersed multipotent stem cells, which can differentiate into nematocytes (stinging cells), gametes, and secretory and nerve cells [48,49]. The *Hydra* epithelium is covered by a multilayered glycocalyx that forms a physicochemical barrier against pathogens [47]. Although *Hydra* lacks mobile phagocytes, stationary epithelial cells can phagocytose and destroy pathogens. Additionally, hydra’s epithelium has pattern recognition molecules, based on lectin–carbohydrate interactions, that recognize compounds present in different microorganisms, and it produces potent antimicrobial peptides (AMPs), which fight pathogens. The *Hydra* genome (which has only ~20,000 genes) lacks conventional Toll-like receptors (*TLRs*), and the production of AMPs is induced by the interaction between proteins containing leucine-rich repeats (LRRs) with proteins lacking LRRS but containing the Toll/Interleukin-1 Receptor (TIR) domain [45]. The expression of AMPs is also controlled by the forkhead box O (FOXO) transcription factor [47,50].

The more advanced invertebrates have specialized immune cells that are similar in morphology and functions to vertebrate macrophages, but different taxonomic groups and species have different types of immune cells. Arthropods and mollusks have open (lacking the vasculature) circulatory systems, and the body cavity is filled with hemolymph, which is analogous to vertebrate blood. Hemolymph usually does not contain red blood cells or hemoglobin (although the intercellular hemoglobin can be present in various insect tissues such as tracheal system and fat body in *Drosophila)* but contains hormones, nutrients, ions, carbohydrates, lipids, pigments, and various types of immune cells (hemocytes). Single-cell RNA sequencing showed that the hematopoietic organ (lymph gland) of *Drosophila* larvae contains several types of myeloid cells: stem-like prohemocytes, intermediate prohemocytes, prohemocytes, plasmatocytes, lamellocytes, adipohemocytes, and crystal cells [51]. Plasmatocytes are very similar to vertebrate macrophages; they phagocytose, remodel tissues, mount an immune response, and express genes involved in metabolism, the cell cycle, and antimicrobial responses [52]. Lamellocytes, which in healthy insects are rare, form in response to parasitic wasps and some environmental factors. They encapsulate pathogens that are too large to be phagocytosed. Adipohemocytes play a role in lipid metabolism, and crystal cells contain a prophenoloxidase (ProPO) enzyme that participates in the melanization immune response (see below) [51,53,54]. Interestingly, as shown in *Tenebrio*, *Manduca sexta*, and *Drosophila*, the activation of the ProPO cascade and melanization pathway shares serine proteases with the Toll pathway [55,56,57]. Studies using single-cell RNA sequencing (scRNA-seq) identified subsets within lamellocytes that express fibroblast growth factor (FGF) receptor breathless and crystal cells that express FGF ligand branchless, which play a role in the response against parasitic wasps [52]. A comprehensive list of genes expressed by *Drosophila* hemocyte subtypes was recently published in [58]. A recent cross-species comparison of single-cell RNA-sequencing data showed that *Drosophila* plasmocytes are homologous to vertebrate monocytes and macrophages and prohemocyte 1 (PH1) is homologous to vertebrate hematopoietic stem cells. Additionally, a subpopulation of *Drosophila* prohemocytes with hematopoietic features was identified as a homolog of vertebrate hematopoietic progenitors [59]. Single-cell RNA sequencing studies of immune cells in shrimp identified three major types of hemocytes: prohemocytes, monocytic hemocytes, and granulocytes. Monocytic hemocytes are like vertebrate phagocytotic macrophages and express specific markers, such as apoptotic and inflammatory-response-related genes: the nucleotide-binding domain and leucine-rich repeat protein 3 (NLRP3) *Nlrp3* and *Casp1* [59]. Another study on shrimp hemocytes revealed that they also share some features with mammalian immune cells. Li et al. [60] analyzed the transcription factor FOXO in shrimp *Marsupenaeus japonicus* challenged with a Gram-negative bacteria *Vibrio anguillarum*. FOXO plays a role in mucosal immune responses in mammals, and it plays a role in gut humoral immune response and low-oxygen immune-like response in invertebrates [61,62,63]. In shrimp, FOXO maintains hemolymph and intestinal microbiota homeostasis by inducing the expression of transcription factor Relish, which belongs to the immune deficiency (IMD) pathway involved in the expression of antimicrobial peptides. Bacteria-challenged shrimps activated FOXO and induced its translocation to the nucleus, where it regulated antibacterial Amp and Relish genes. Additionally, FOXO induced phagocytosis via the upregulation of the phagocytotic receptor—the scavenger receptor C (Src)—and Rab5 and Rab7, two small GTPases regulating phagosome trafficking to the lysosomes [61].

There are also many studies analyzing the immune cells in colonial tunicates, a group of marine invertebrate chordates that are the closest living ancestors of vertebrates. Their circulatory system consists of the heart and extracorporeal vasculature, which share the blood supply within the colony [64]. Studies of colonial tunicate *Botryllus schlosseri* showed that their hematopoietic system has a mixture of invertebrate and vertebrate features. Rosental et al. [65] identified three subpopulations of phagocytic cells: major phagocytic amoebocytes; large phagocytes, which mainly contributed to allogeneic phagocytosis; and myeloid cells, which were the main contributors to phagocytosis. They also identified large granular lymphocyte-like (LGL) cells (also called the morula cells) with cytotoxic properties, and they were similar in morphology and function to mammalian natural killer (NK) cells. Despite similarities to mammalian lymphocytes, these cells mainly express tunicate-specific genes [65]. 

Another type of invertebrate innate immune cells is multinucleated giant hemocytes (MGHs), which fight parasitoid infections (such as wasp eggs and larvae) in insects [66]. Invertebrate MGHs are probably an evolutionary precursor of macrophage-derived multinucleated giant cells—a hallmark of chronic inflammation, aging, and cancer in vertebrates [67]. MGHs circulating in insect hemolymph are highly mobile, and when they encounter the parasite, they encapsulate and kill it. The multinuclearity of MGHs serves to amplify the expression of anti-parasitic genes. The ultrastructural and live cell imaging of *Drosophila ananassae* infected with parasitoid wasps showed that MGHs form an intricate system of intracytoplasmic membrane structures and exosomes, which are used for parasite encapsulation. MGHs also express high levels of hemolysin-like proteins and pore-forming toxins of prokaryotic origin, which are used for parasite elimination [66]. Studies on *Anopheles gambiae* showed that in response to *Plasmodium* infection, its hemocytes differentiate via a homolog of the *Drosophila* Dorsal Rel 1 signaling pathway, into giant cells (megacytes), which are recruited to the midgut surface where they eliminate *Plasmodium* ookinetes [68].

## 4. Melanization Immune Response

In many arthropods, including insects such as *Drosophila* and mosquitoes, the immune response against invading pathogens involves the melanization process, which is when the host produces melanin that encapsulates, isolates, prevents dissemination, and kills the pathogen (Figure 2). Because melanization occurs within a few minutes after infection, it is a much faster response than taking several hours to fully mobilize and activate immune cells/effectors [69,70]. Melanin is a generic name for a group of natural pigments found in most organisms that are produced in the process of melanogenesis, which involves the oxidation of phenols to quinones followed by their polymerization [53,70,71,72,73]. In healthy organisms, melanin is produced and stored in specialized cells—melanocytes, which besides making melanin secrete signaling molecules, such as cytokines, POMC peptides (the source of adrenocorticotropic hormone (ACTH), β-lipotropin, β-endorphin, and neuropeptides), catecholamines, and nitric oxide [74]. Upon infection, the pattern recognition receptors on host cells induce a cascade of serine proteases, which activate phenol oxidase (PO), an enzyme directly involved in melanin biosynthesis. PO catalyzes the oxidation of phenols to quinones, which cross-link to form melanin over the pathogen’s surface. Melanin is either produced directly from circulating precursors, or melanin-producing cells adhere tightly to each other and surround the pathogen, forming a tight capsule around it, and, in the next step, they deposit melanin [57]. Because the quinones and reactive oxygen species produced during melanization are toxic, melanization, besides sequestering the pathogen, also kills it. The toxicity of melanization intermediates requires the tight regulation of the melanization process to prevent damage to the host. Thus, the organisms contain PO pathway inhibitors, such as serpins and pacifastins. Genetic studies in *Drosophila* showed that the mutations of melanization inhibitors (serpins) cause spontaneous melanization and lethality [72,75]. The enzymes and pathways involved in the melanization process are described in detail in [70].

## 5. Innate Immune Memory

In the most basic definition, immune memory is the ability of a quiescent immune system to recall and respond to challenges encountered in the past, resulting either in the development of tolerance or the potentiation of an immune response. Since its discovery in 1997 in B cells, immune memory has been, for many years, attributed only to adaptive immunity [76]. However, recent studies showed that mouse macrophages have three types of immunological memory: innate nonspecific immunological memory; mediated by the pattern recognition receptor (PRR); innate antigen-specific immunological memory, which is dependent on the macrophage MHC-I receptor (the paired immunoglobulin-like receptor-A); and adaptive immunological memory, created with assistance from various subpopulations of adaptive immune cells, such as T and B cells [77,78,79,80,81]. In vitro studies and mice lacking B and T cells showed that vaccination with *Bacillus Calmette-Guérin* (BCG) resulted in immune priming, the development of trained immunity, and cross-protection against non-mycobacterial infections, which involved the epigenetic reprogramming of monocytes and macrophages (see below). For three months after vaccination, these innate immune cells upregulated inflammatory factors and reactive oxygen species and increased microbicidal and phagocytic responses after a second exposure to the same or similar pathogen [82,83,84,85,86,87,88]. Because innate immunity evolved before the emergence of adaptive immunity, the innate immune memory of invertebrates must thus be a prototype for adaptive immune memory. In *Drosophila melanogaster*, after the uptake of viral RNA, the hemocytes produce anti-viral short interfering RNAs (siRNAs), which, after being secreted in exosomes, act like vertebrate antibodies by conferring protection against the virus in naive flies. In *Aedes aegypti*, viral DNA integrated into the host genome is transcribed into PIWI (P-element induced wimpy testis in *Drosophila)*-interacting RNAs (piRNAs), which confer heritable protection on next generations against subsequent encounters with the virus [79,85,89,90,91]. Even plants that do not have motile immune cells have immune memory, the so-called systemic acquired resistance (SAR), which can last for a lifetime and depends on salicylic acid (SA) signaling and heritable epigenetic changes in the plant genome [92,93,94,95,96].

## 6. Epigenetics of Invertebrate Immunity

In the broadest sense, epigenetics refers to heritable traits that do not depend on changes in the DNA sequence (Figure 3). The main categories of epigenetic modifications are those changing the structure and transcriptional accessibility of chromatin, DNA methylation, histone modifications, noncoding RNAs, and RNA epigenetics that consist of post-transcriptional modifications of tRNA, mRNA, and rRNA, which modulate gene expression [97,98,99]. Another recently discovered category is the so-called RNA recoding, which is an epigenetic process of RNA editing that alters the amino acid sequence of proteins [100,101,102,103]. Epigenetic changes can be induced not only by environmental factors but also by pathogens, which train and boost innate immune responses but also hijack epigenetic mechanisms for their benefit to subdue immunity [98]. Although some of the epigenetic modifications are rapidly reversible, others can be long-lived, persisting not only through many generations of mitotically dividing cells but also by being passed on through germ cells from parents to the offspring, contributing to the development of heritable innate immune memory [103,104].

In invertebrates and vertebrates, chromatin architecture and changes from open (accessible for the transcription factors) to condensed (transcriptionally silent) configurations are regulated by reversible histone acetylation or methylation [104]. The reversible acetylation of histones depends on histone acetyltransferases (HATs), which transfer the acetyl group from acetyl-CoA to the lysine amino acids of histone tails and histone deacetylases (HDACs), which remove the acetyl groups [105]. The dysregulation of HAT/HDAC affects invertebrate antimicrobial immune response. In honeybees, the downregulation of HDACs upregulates histone acetylation, the Janus kinase/signal transducer and activator of the transcription (JAK/STAT) pathway, and the expression of antimicrobial peptides (AMPs) [106]. In mosquitoes, viral infection induces the expression of histone acetyltransferase CBP, opens chromatin structure, and switches on immune genes, while its downregulation increases viral titer and mosquito mortality [107]. Also, insect parasitoids modulate the expression of HAT/HDAC genes and suppress host immunity [108]. Epigenetic modifications are heritable: for example, heat stress changes histone acetylation patterns and induces the epigenetic transgenerational inheritance of heat resistance in the parthenogenetic brine shrimp model [101,109].

Another epigenetic modification that regulates gene expression is DNA methylation catalyzed by DNA methyltransferases (DMMTs) [110]. Although average global DNA methylation in invertebrates is lower (20–40%) than in vertebrates (80%), in some invertebrate species, such as prawns, only 2% of the genome is methylated, while in other species, such as clam worms, 80% of the genome is methylated [111]. Although it is well documented that changes in methylation patterns can lead to immune disorders in humans [112], much less is known about its effect on invertebrate immunity. Recent studies showed that oyster hemocytes express a high level of DNMT3 transferase, and the bacterial challenge upregulates the expression of *DNMT3 mRNA* [113], while the treatment with epigenetic modifying drug decitabine (DNA-specific methyltransferases inhibitor) upregulates immune-related genes [114]. It seems that, at least in marine invertebrates, there is a tendency to increase DNA methylation in response to heat stress and other environmental challenges [115,116]. In insects, such as silkworms, the suppression of DNMTs increases the apoptosis of virus-infected cells and suppresses virus proliferation [117]. The DNMT inhibitor (5-AZA) upregulates the expression of antimicrobial peptides (AMPs) and inhibits bacterial replication in silkworms [118]. A similar correlation between DNA methylation and the expression of AMPs was also described in other insects [119,120,121]. Recent studies indicate that epigenetic changes responsible for the development of intragenerational or intergenerational innate immune memory in invertebrates may depend not only on DNA but also on RNA modifications and/or recoding [100,122]. In invertebrates and vertebrates, the adenosine deaminases acting on RNA (ADARs) edit RNAs by changing the adenosine to inosine (A-to-I). mRNA editing within the coding region results in recoding (codon changes), which, in turn, produces protein variants with a changed composition of amino acids and creates phenotypic plasticity [123,124]. Thus, it is possible that RNA recoding may produce protein variants of different functionality not only in response to environmental factors but also to pathogens. Although RNA recoding is a rare process in humans, in some invertebrates, such as cephalopods (squid, cuttlefish, and octopus), around 60% of all mRNAs are recoded [124,125]. Besides adenosine-to-inosine (A-to-I) RNA recoding, other modifications of mRNA, rRNA, tRNA, snRNA, and miRNA, such as N^6^-methyladenosine (m^6^A), 5-methylcytosine (m^5^C), N^1^-methyladenosine (m^1^A), N^7^-methylguanosine (m^7^G), N^4^-acetylcytosine (ac^4^C), pseudouridine (Ψ), and uridylation, modulate immune responses by regulating immune cell polarization, activation, migration, and differentiation [120]. There are many examples of RNA modification/recoding affecting macrophage and monocyte functions in mammals. For example, m^6^A affects the antiviral and anticancer response of monocytes and macrophages; directs, through the Foxo-1-IL-10 axis, macrophage polarization toward M2 phenotype; and promotes macrophage pyroptosis [122,126,127]. Studies of the DNA and RNA methylation status in the beetle *Tenebrio molitor* showed the presence of methylated cytosine residues (5mC) in RNA during immune priming after a second challenge with bacteria and fungus [128], which suggests that RNA modification/recoding may also affect innate immune memory in invertebrates.

## 7. Intergenerational and Transgenerational Immunity

In many animal and plant species, the memory of an encounter with pathogens can be transmitted to the next generation (intergenerational immunity) or several subsequent generations (transgenerational immunity, transgenerational immune priming, and TGIP) [129,130,131,132]. The transmission of short-lived immunity, based on non-genetic factors, between the mother and offspring, is well known in mammals, where the antibodies developed by the mother pass through the placenta and are transferred via breast milk to the newborn, and this also occurs in other vertebrates and invertebrates whose eggs may contain fragments of pathogens, maternal antibodies, antimicrobial agents, or silencing RNAs [130,131,133,134]. Interestingly, in animals with male pregnancy, such as seahorses, immunity can be passed to the egg from the father [135]. However, as intergenerational and transgenerational immunity must be long-lived, it requires a more permanent solution, i.e., an immune inheritance. TGIP has been described in many species of Crustacea, Coleoptera, Hymenoptera, Orthoptera, and mollusks [130]. Studies in the worm *C. elegans* showed that the immune memory of pathogen encounters transferred between generations regulates immune gene expression levels and imprints protective behavior with respect to pathogen avoidance in the progeny [132]. For example, parental exposure to *Pseudomonas vranovensis* induces the upregulation of cysteine synthase genes *cysl-1* and *cysl-2* and *rhy-1* [136]. Cysteine and other amino acid metabolism and *rhy-1*(which encodes a multi-pass transmembrane protein and inhibits a hypoxia inducible factor in *C. elegans*) are known to promote immune cell functions [137,138]. Some studies suggested the involvement of RNAi in the establishment of transgenerational immunity in *C. elegans* after virus exposure, but other studies did not confirm the involvement of antiviral interfering RNAs in this process [139,140,141]. 

A global proteome analysis of mealworm beetle *Tenebrio molitor* eggs from mothers primed with Gram-positive and -negative bacteria showed the abundance of immunologically active proteins and peptides, such as heat shock proteins; annexin; prophenoloxidase; transferins; perilipin; tenecins 1, 2, and 4; defensin; coleoptericin; and attacin, but no RNAs were found [131]. Although this study clearly showed that these immune factors protected *Tenebrio* eggs from infection, it is still unknown what mechanisms (most probably epigenetic, see below) are involved in the establishment of transgenerational immunity in this species. Studies in another model insect, house fly *Musca domestica,* showed that adults primed with heat-killed *C. albicans* can transfer immunity to the next generations. A transcriptome analysis of primed adults identified 24 upregulated and 6 downregulated genes, including metabolic genes and Toll signaling and phagosome pathways, and 154 differentially expressed (80 upregulated and 74 downregulated) genes were identified in their larvae when exposed to a lethal dose of *C. albicans* [142]. These results indicate that the immune response of the parent induces differential gene expression in the offspring, but again, there is no information on how this happens and what the exact mechanisms underlying the emergence of transgenerational immunity are. 

The analysis of transgenerational immunity is complicated not only by differences in pathogen virulence but also by the longevity and habituation mode (philopatry versus dispersal) of different host species. By analyzing published data on 21 invertebrate species, Pigeault et al. [129] developed a theoretical host–parasite model to understand how the above parameters affect the evolution of transgenerational immunity. One of the conclusions is that transgenerational immunity will only develop when the mother is exposed to a moderately virulent pathogen. If virulence is too high, the chances of survival and reproduction of the individual are so low that there is no reason to invest in the maternal transfer of immunity. Similarly, short-lived species would not benefit from transgenerational immunity because they naturally die very fast, and the chances of encountering the same pathogen they are immune to are minuscule. There is also a strong correlation with the dispersal behavior. When the species changes location, the chance of encountering the same pathogen the mother was exposed to is so low that there is no benefit in having mother-derived immunity. In summary, only long-lived and philopatric species have an incentive to develop transferable immune memory [129].

## 8. Mechanisms of Self and Non-Self-Recognition, Natural Transplantation, and Germ and Somatic Cell Parasitism

The ability to recognize itself and genetically similar individuals is not only indispensable for self-preservation but also for sexual reproduction and the resulting exchange of genetic material between mating partners. In jawed vertebrates, including mammals, the main molecules responsible for allorecognition are the major histocompatibility complex (MHC) molecules, which present self- or non-self-antigens to the T cell receptors. The question is if there are any orthologs of MHC in the invertebrates and, even more generally, if the mechanisms of allorecognition in invertebrates are related to those in vertebrates [143]. Excellent models for studying the mechanisms of allorecognition in invertebrates are the compound organisms (colonies) of asexually propagating polyps of cnidarians, such as *Hydractinia,* and the zooids of colonial ascidians, such as *Botryllus.* When an expanding colony touches a different but genetically identical colony, they permanently fuse. Such a process of natural transplantation may have various outcomes. If the encountered colony is only partially compatible, they fuse partially (fuse for a short time and separate or cyclically fuse and reject), while meeting with a genetically non-compatible (allogeneic) colony leads to autophagy, necrosis, and rejection [143,144,145,146].

In *Botryllus,* genetic compatibility depends on a polymorphic locus *fuhc* [147], which consists of two independent genes with hundreds of alleles, separated by a 227 bp intergenic region. One gene encodes a secreted form of protein, and another encodes a membrane-bound form. These proteins contain immunoglobulin (Ig) domains that are not related to vertebrate MHC [147]. Another polymorphic locus *fester*, separated by ~300 kb from *fuhc,* codes for activating and inhibiting receptors for histocompatibility reactions and plays a role in both initiating the reaction and discriminating between *fuhc* alleles [148,149]. Additionally, a related gene, called the *uncle fester*, which is co-expressed with *fester*, is required for incompatible but not compatible reactions [148].

Although cnidarians do not have immune cells, some can discriminate between self- and non-self-antigens. Colonial cnidarians have two histocompatibility loci (*alr1* and *alr2*), which encode membrane-bound proteins. Despite this, they also contain immunoglobulin-like domains that are not homologous to those in *Botryllus*. Detailed analyses of the Hydractinia genome showed that the alr2 locus contains the coding sequence for the CDS7 protein, which is a transmembrane receptor with three extracellular immunoglobulin Ig-like domains [147,150,151]. *Hydractinia* polyps will fuse when they are genetically identical or share one allele at both loci (*rr/rr × rr/rf*), while the different alleles (*rr/rr × ff/ff*) result in rejection and tissue necrosis. It seems that mechanisms of allorecognition are highly variable between different species and groups of invertebrates. For example, in demosponges such as *Amphimedon,* allorecognition is controlled by a cluster of aggregation-related genes. They encode proteoglycan aggregation factors (containing an extracellular Calx-β domain), which besides allorecognition also control cell adhesion [152]. In some invertebrates, such as solitary freshwater cnidarian *Hydra*, allorecognition does not lead to rejection [153], suggesting that solitary cnidarians do not have self-/non-self-recognition systems present in colonial cnidarian species. 

Another fascinating issue is the fate of genetically different cell lineages within a fused organism. The fusion of conspecific organisms may lead to chimerism and genetic heterogeneity, where the somatic and germ cells of two different genotypes coexist, or to a competition resulting in somatic or germ cell parasitism, or a combination of these processes [154,155]. In addition, there is also a phenomenon of allogeneic/chimeric resorption when, after successful fusion, one partner undergoes massive phagocytosis and is resorbed by the other [145]. Such a resorption process is genetically controlled by the histocompatibility alleles. Studies showed that in *Botryllus schloserri*, allogeneic resorption depends on the *fuhc* locus and resorption/histocompatibility loci *rehc1* and *rehc2*, which represent three levels of resorption screening, with the general rule that the more heterozygotic partner will resorb the more homozygotic one. The first compatibility screening occurs at the fuhc level, resulting in the resorption of a more homozygous partner. If the resorption cannot be determined at the fuhc level, it will be dictated by heterozygosity/homozygosity at *rehc1*, and subsequently, if this fails, it will be dictated by *rehc2*. Interestingly, quite often, although the eliminated partner is not genetically represented in the soma, its germline cells survive and partially or completely take over the gonads of the winning partner. Such germline parasitism will result in the offspring bearing the genotype of the parasitic germ cells [146]. Studies of the genotype signature using amplified fragment length polymorphisms (AFLPs) as the genetic markers of experimentally created fusion chimeras of juvenile *Botryllus schlosseri* showed that there were several outcomes of fusion in terms of somatic/germline participation [154]. Germline and somatic tissues within a chimera had identical genetic markers, which resulted from complete stem cell parasitism in germline and somatic tissues or the coexistence of both genotypes. Germline matched one of the somatic signatures, indicating germline parasitism and the coexistence of somatic genotypes. All germline cells had the same genotype that was different from the somatic genotype, and finally, all somatic cells had the same genotype, but the germline was chimeric, indicating somatic parasitism or resorption dominance but genotype coexistence in the germline [154]. Because these studies were performed on juveniles, which have abundant stem cell populations, further studies are needed to establish the participation of stem cells in juvenile versus adult fusion, what the mechanisms of these different post-fusion outcomes are, and their advantages/disadvantages for environmental fitness and survival.

## 9. Defense against Transposons

The immune defense operates not only at the organismal level but also at the level of the genome. Genomic immunity is necessary to protect against selfish transposable parasitic DNAs (transposable elements, TEs), which, in contrast to regular pathogens, do not bear typical antigens that are readily recognizable by the immune system. Additionally, these selfish DNAs must be somehow distinguished from the self-DNA. Because TEs can replicate and insert themselves into various locations within the genome, they play a positive role in the evolution of genomes but also pose a significant threat by promoting DNA breaks and spurious recombination. As preserving genomic integrity is especially important for germ-line cells, they developed a sophisticated defense mechanism against TEs and other potentially detrimental foreign nucleic acids [156]. The germ cells have a special RNA–protein complex containing guide-dependent Argonaute nucleases belonging to the PIWI (P-element induced wimpy testis in *Drosophila*) subfamily and small (24–31 nucleotides in length), non-coding piRNA (PIWI protein binding RNA). The piRNA pathway silences TEs via two complementary mechanisms: suppressive alterations of chromatin around the targeted genes and cleaving TE transcripts in the cytoplasm [157,158,159,160,161]. The piRNA pathway operates in all animals but is best described in insect and mouse germ cells [161,162,163,164]. piRNAs are either transcribed from the TE copies or the clusters (200 kilobases long in flies) located in the pericentromeric and sub-telomeric regions of chromosomes. The piRNA transcriptional clusters are highly enriched in the dysfunctional remnants of TEs and, by serving as a molecular memory of TE invasion, are the basis of immune defense against TEs [161,165,166]. Transcription from piRNA clusters generates antisense sequences relative to TE mRNAs. Their ultimate role is to find complementary TE mRNAs and guide them toward PIWI proteins, which cleave and destroy them. After transcription, piRNA cluster transcripts are transported to the cytoplasm where they are processed into piRNA intermediates. Subsequently, piRNA intermediates bind PIWI proteins, and after further modification (usually methylation and trimming), mature functional PIWI-piRNA complexes are ready to destroy the invading TEs [161,167].

In *C. elegans,* the piRNA pathway collaborates with other small RNA pathways to silence the invading TEs and create a trans-generational memory that protects self-mRNAs but allows piRNAs to recognize, without prior exposure, foreign sequences [168,169]. The transcriptome of *C. elegans* gonads generates a population of diverse small RNAs called 22G- RNAs, which are associated with Argonaute CSR-1 (chromosome segregation and RNAi deficient) proteins. The 22G-RNA/CSR1 complexes are transmitted to the progeny where they recognize the self (parental) transcripts and protect them from destruction by the piRNA pathway, which in *C. elegans* consist of PIWI family proteins PRG1/WAGOs and the 21U-RNA in piRNA [161,170,171,172]. Recent studies identified a novel protein, PID-5, which plays a role in RNA-induced epigenetic silencing in the *C. elegans* embryo. It regulates proteolytic cleavage in the silencing pathway and facilitates the balanced production of 22G-RNA signals for transgenerational silencing [171]. The analogous mechanisms of genomic self-recognition and protection against transposable sequences, based on siRNAs and RNA binding proteins, operate in most unicellular and multicellular organisms and progenitor cells [161,173,174,175,176].

## 10. Conclusions

The invertebrate models give an insight into the functioning and evolution of the innate immune system and help define conserved features of innate immunity. They also provide invaluable information on short and long immune memory and the possible mechanisms of transgenerational immunity. The absence of cellular and humoral components of adaptive immunity makes invertebrates a perfect model for studying how sensu stricto innate immunity functions in isolation without being influenced by an alliance with adaptive immunity. Additionally, invertebrate models allow the discovery of novel pathogens and antimicrobial agents. Analyses of immune responses in invertebrate models may be a powerful tool for the fast and low-cost screening of the immunomodulatory effects of various environmental factors and for exploring the interactions between the host and microbiome. Further studies in all the above-described areas are necessary to fully understand the scope of invertebrate immunity, make valid conclusions and comparisons between invertebrate and vertebrate immune cells and processes, and shed light on the evolution and development of different branches of immunity.

Finally, we want to draw the readers’ attention to the emerging field of immunometabolism. Studies of the last decade have indicated that extracellular signals, including those derived from pathogens, affect the uptake and catabolism of nutrients in immune cells and have shown how metabolic pathways shape immune cells’ fate, phenotypes, and effector functions [177,178,179].

## Figures and Tables

**Figure 1 ijms-25-01072-f001:**
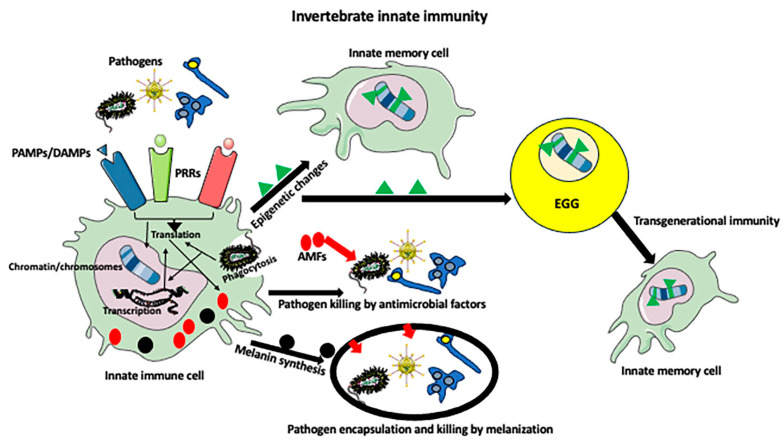
Invertebrate innate immunity. PAMPs (pathogen-related molecular patterns) and DAMPs (damage-associated molecular patterns) originating from the pathogen invasion are recognized by the PPRs (pattern recognition receptors) on innate immune cells. Recognition of pathogen antigens and pathogen phagocytosis activates the immune call and induces a cascade of signaling pathways, which affect protein expression at translational and transcriptional levels. Activated immune cells produce antimicrobial factors (AMFs), such as antimicrobial peptides, oxyradicals, and melanin, which destroy pathogens. The metabolic changes in the activated immune cell can also result in the epigenetic modifications of chromatin and chromosomes. The epigenetic modifications of the genome will result in the imprinting of pathogen-encounter memory and the formation of innate memory cells. When pathogen-encounter-induced epigenetic modifications occur in gametes (usually in the egg, but sometimes also in sperm) they are transmitted to the next generation(s) where they program the innate immune cells of the offspring for rapid and strong anti-pathogen responses.

**Figure 2 ijms-25-01072-f002:**
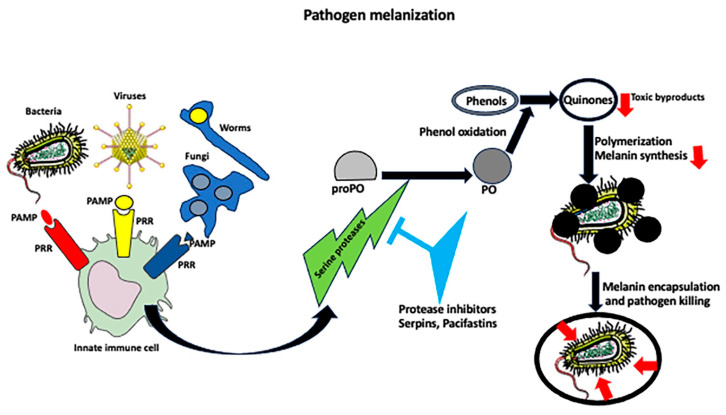
Pathogen melanization pathway. Recognition of pathogen antigens by immune cell receptors switches on the melanin synthesis pathway. In this pathway, the activated serine proteases convert an inactive phenol oxidase enzyme (prePO) into an active phenol oxidase (PO). The PO catalyzes the oxidation of phenols to quinones, which polymerize into melanin that is deposited on the pathogen’s surface. Eventually, a pathogen is encapsulated by the layer of melanin and killed by quinones, and reactive oxygen species are produced during the melanization process.

**Figure 3 ijms-25-01072-f003:**
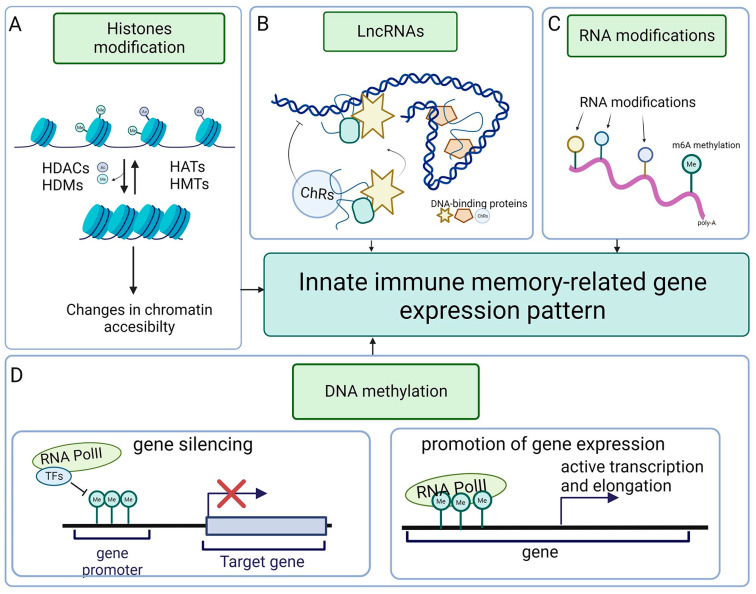
Overview of epigenetic modifications in immune-related gene expression in invertebrates. (**A**) Reversible histone acetylation and methylation are mediated by HATs/HDACs and HDMs/HMTs, respectively, leading to changes in chromatin structure. Histone acetylation/methylation opens chromatin structure and enables easier access to transcription machinery. Histone deacetylation/demethylation has the opposite effect, repressing gene transcription by tightening the chromatin structure. (**B**) LncRNAs recognize and recruit epigenetic modifiers (DNA-binding protein complexes) onto specific loci or prevent the complexes from binding (right) and introduce changes in the 3D chromatin structure. (**C**) RNA undergoes modification by adding chemical groups. The most common modification is M6-methyladenosine methylation. (**D**) DNA methylation leads to gene transcription silencing when DNA undergoes methylation at the promoter level (left panel), which blocks access to transcription factors. In contrast, the gene body part’s methylation (right panel) promotes gene expression and supports active transcription and gene elongation.

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
