# Peer review of "Invertebrate Immunity, Natural Transplantation Immunity, Somatic and Germ Cell Parasitism, and Transposon Defense"

_ijms, 2024, doi:10.3390/ijms25021072_

Round 1
Reviewer 1 Report
Comments and Suggestions for Authors
I have reviewed the manuscript titled "Invertebrate Immunity: Natural Transplantation Immunity, Somatic and Germ Cell Parasitism, and Transposon Defense" by Malgorzata Kloc et al. The authors have undertaken the challenging task of summarizing and generalizing a broad and complex topic such as invertebrate immunity. I appreciate the difficulty of this endeavor and understand the necessity of selecting specific topics that the authors deem most pertinent. However, the authors should try to broaden the models they reference across different taxa. They should also try to identify recent trends in the literature and point to their view of the direction the scientific progress is going. This will help to differentiate their work from other past reviews on the topic.
Updating Classical Views (Ln 32-34): The traditional perspective of innate and adaptive immunity presented needs to be updated to align with recent scientific advancements.
Inclusion of Humoral Immunity (Ln 34-38): The omission of humoral immunity in the initial sections, despite its later discussion, requires rectification for continuity.
Examples of High Affinity Ligand-Receptors (Ln 75): Specific examples of TLRs with high affinity to PAMPs would substantiate the claims made in this section.
Expanding on Antimicrobial Defense Mechanisms (Ln 121-123): Besides ETosis, incorporating details about autophagy and proteasome formation would present a more comprehensive view of antimicrobial defenses.
Broader Taxonomic Examples: While the focus on certain taxa is noted, the manuscript would benefit from more diverse examples across different taxa to provide a holistic view of invertebrate immunity.
Role of of Organs and Tissues in Immunity (Ln 138-143): The discussion predominantly centers on hydra, while the immune roles of organs like the gut and fat body in other animals are overlooked.
Diversity and Comparison of Immune Cells (Ln 147-148): A deeper exploration of whether distinct immune cell types exist across taxa or if terminological differences arise from independent descriptions would be insightful.
Recent Research and Developments (Ln 152-155, 196-197): Incorporating recent studies, such as [doi.org/10.1371/journal.pgen.1011077] and findings on giant hemocytes [10.7554/eLife.81116], would enrich the manuscript.
Relevance and Integration of Section 5: The necessity and placement of Section 5 within the manuscript should be re-evaluated for better coherence.
Incorporating Established Concepts (Ln 325-327): The manuscript should utilize well-established concepts like trained immunity and priming for consistency.
Hemocyte-Dependent Priming Mechanisms (Ln 332-340): Additional information on hemocyte-dependent priming, as discussed in various reviews, would provide a more comprehensive understanding.
Discussion on Immunometabolism: The manuscript could benefit from a discussion on the role of immunometabolism in shaping innate immunity, which is a significant aspect of the field.
Comments on the Quality of English LanguageOnly minor issues were detected. This should be easily detected upon editorial proof reading.
Author Response
Review 1
I have reviewed the manuscript titled "Invertebrate Immunity: Natural Transplantation Immunity, Somatic and Germ Cell Parasitism, and Transposon Defense" by Malgorzata Kloc et al. The authors have undertaken the challenging task of summarizing and generalizing a broad and complex topic such as invertebrate immunity. I appreciate the difficulty of this endeavor and understand the necessity of selecting specific topics that the authors deem most pertinent. However, the authors should try to broaden the models they reference across different taxa. They should also try to identify recent trends in the literature and point to their view of the direction the scientific progress is going. This will help to differentiate their work from other past reviews on the topic.
1.Updating Classical Views (Ln 32-34): The traditional perspective of innate and adaptive immunity presented needs to be updated to align with recent scientific advancements.
Response: As requested, we updated the traditional view and added the reference
Černý J, Stříž I. Adaptive innate immunity or innate adaptive immunity? Clin Sci (Lond). 2019 Jul 17;133(14):1549-1565. doi: 10.1042/CS20180548.
Zhu Q, Zhang M, Shi M, Liu Y, Zhao Q, Wang W, Zhang G, Yang L, Zhi J, Zhang L, Hu G, Chen P, Yang Y, Dai W, Liu T, He Y, Feng G, Zhao G. Human B cells have an active phagocytic capability and undergo immune activation upon phagocytosis of Mycobacterium tuberculosis. Immunobiology. 2016 Apr;221(4):558-67. doi: 10.1016/j.imbio.2015.12.003.
Stögerer T, Stäger S. Innate Immune Sensing by Cells of the Adaptive Immune System. Front Immunol. 2020 May 29;11:1081. doi: 10.3389/fimmu.2020.01081.
- Inclusion of Humoral Immunity (Ln 34-38): The omission of humoral immunity in the initial sections, despite its later discussion, requires rectification for continuity.
Response: As requested, we added info and references on humoral immunity
Mantovani A, Garlanda C. Humoral Innate Immunity and Acute-Phase Proteins. N Engl J Med. 2023 Feb 2;388(5):439-452. doi: 10.1056/NEJMra2206346.
Kanellopoulos JM, Ojcius DM. Development of humoral immunity. Biomed J. 2019 Aug;42(4):207-208. doi: 10.1016/j.bj.2019.08.003.
Foo SS, Reading PC, Jaillon S, Mantovani A, Mahalingam S. Pentraxins and Collectins: Friend or Foe during Pathogen Invasion? Trends Microbiol. 2015 Dec;23(12):799-811. doi: 10.1016/j.tim.2015.09.006.
Zhang XL, Ali MA. Ficolins: structure, function and associated diseases. Adv Exp Med Biol. 2008;632:105-15. PMID: 19025118.
- Examples of High Affinity Ligand-Receptors (Ln 75): Specific examples of TLRs with high affinity to PAMPs would substantiate the claims made in this section.
Response: As requested, we added the examples and references
Werling D, Jungi TW. TOLL-like receptors linking innate and adaptive immune response. Vet Immunol Immunopathol. 2003 Jan 10;91(1):1-12. doi: 10.1016/s0165-2427(02)00228-3.
- Expanding on Antimicrobial Defense Mechanisms (Ln 121-123):Besides ETosis, incorporating details about autophagy and proteasome formation would present a more comprehensive view of antimicrobial defenses.
Response: As requested, we added information and relevant references on autophagy and proteasome
Nandi D, Tahiliani P, Kumar A, Chandu D. The ubiquitin-proteasome system. J Biosci. 2006 Mar;31(1):137-55. doi: 10.1007/BF02705243.
Thomas JH. Adaptive evolution in two large families of ubiquitin-ligase adapters in nematodes and plants. Genome Res. 2006 Aug;16(8):1017-30. doi: 10.1101/gr.5089806.
Cui B, Lin H, Yu J, Yu J, Hu Z. Autophagy and the Immune Response. Adv Exp Med Biol. 2019;1206:595-634. doi: 10.1007/978-981-15-0602-4_27.
Kuo CJ, Hansen M, Troemel E. Autophagy and innate immunity: Insights from invertebrate model organisms. Autophagy. 2018;14(2):233-242. doi: 10.1080/15548627.2017.1389824.
- Broader Taxonomic Examples: While the focus on certain taxa is noted, the manuscript would benefit from more diverse examples across different taxa to provide a holistic view of invertebrate immunity.
Response: Our review covers, on purpose, too many topics to be a holistic representation of all invertebrate taxa. Instead of a comprehensive description of all invertebrate taxa, we intended to point out the most interesting, in our opinion, topics for further reading. In this context, we think that exemplifying additional taxa is not necessary.
- Role of Organs and Tissues in Immunity (Ln 138-143): The discussion predominantly centers on hydra, while the immune roles of organs like the gut and fat body in other animals are overlooked.
Response: The Hydra was chosen on purpose. The whole point of this description was that Hydra lacks mobile phagocytes and does not have organs or tissues beyond the epithelial layers, but still has an immune response. So, mentioning here the adipose tissue or gut would not be appropriate.
- Diversity and Comparison of Immune Cells (Ln 147-148): A deeper exploration of whether distinct immune cell types exist across taxa or if terminological differences arise from independent descriptions would be insightful.
Response: In the same chapter, below the line 141-148 we describe different types of invertebrate immune cells
- Recent Research and Developments (Ln 152-155, 196-197): Incorporating recent studies, such as [doi.org/10.1371/journal.pgen.1011077] and findings on giant hemocytes [10.7554/eLife.81116], would enrich the manuscript.
Response: We added these citations and incorporated the respective information
Yoon SH, Cho B, Lee D, Kim H, Shim J, Nam JW. Molecular traces of Drosophila hemocytes reveal transcriptomic conservation with vertebrate myeloid cells. PLoS Genet. 2023 Dec 19;19(12):e1011077. doi: 10.1371/journal.pgen.1011077.
Barletta ABF, Saha B, Trisnadi N, Talyuli OAC, Raddi G, Barillas-Mury C. Hemocyte differentiation to the megacyte lineage enhances mosquito immunity against Plasmodium. Elife. 2022 Sep 2;11:e81116. doi: 10.7554/eLife.81116.
- Relevance and Integration of Section 5: The necessity and placement of Section 5 within the manuscript should be re-evaluated for better coherence.
Response: We moved the Section 5 to the end of the manuscript (it is now Section 8)
- Incorporating Established Concepts (Ln 325-327): The manuscript should utilize well-established concepts like trained immunity and priming for consistency.
Response: we modified the text
- Hemocyte-Dependent Priming Mechanisms (Ln 332-340): Additional information on hemocyte-dependent priming, as discussed in various reviews, would provide a more comprehensive understanding.
Response: This information and relevant references were provided in the chapter 2 (Invertebrate immunity)
- Discussion on Immunometabolism: The manuscript could benefit from a discussion on the role of immunometabolism in shaping innate immunity, which is a significant aspect of the field.
Response: Throughout the manuscript, we described several times how the metabolic changes shape innate cell phenotype and response and vice versa how pathogens induce metabolic changes. Although we think that the addition of a whole section on immunometabolism would be beyond the focus of this review, in the Conclusion section we pointed readers to the immunometabolism field and added relevant reference.
Ganeshan K, Chawla A. Metabolic regulation of immune responses. Annu Rev Immunol. 2014;32:609-34. doi: 10.1146/annurev-immunol-032713-120236.
Bahat, A., MacVicar, T., & Langer, T. (2021). Metabolism and Innate Immunity Meet at the Mitochondria. Frontiers in Cell and Developmental Biology, 9, 720490. https://doi.org/10.3389/fcell.2021.720490
Ferreira AV, Domiguéz-Andrés J, Netea MG. The Role of Cell Metabolism in Innate Immune Memory. J Innate Immun. 2022;14(1):42-50. doi: 10.1159/000512280. Epub 2020 Dec 30. Erratum in: J Innate Immun. 2021;13(3):194.

Reviewer 2 Report
Comments and Suggestions for Authors
This review is of some interest since it focus on some recent findings such as transplantation immunity and as such may warrant inclusion in this journal after some revision. For this reviewer though much of recent work on epigenetics for example are still in its infancy and thus the authors could perhaps be a bit more critic in their review to the rather few observations compared to other more solid data on invertebrate immunity. But all in all this type of review is of enough interest to warrant inclusion in a journal. One minor thing is that the activation of the proPO cascade and that of the Toll pathway shares the same proteinases as was first shown in Tenebrio by Bok Luel Lee in 2008-2009 and later basically confirmed in both Manduca sexta and Drosophila.
Author Response
Review 2
This review is of some interest since it focus on some recent findings such as transplantation immunity and as such may warrant inclusion in this journal after some revision. For this reviewer though much of recent work on epigenetics for example are still in its infancy and thus the authors could perhaps be a bit more critic in their review to the rather few observations compared to other more solid data on invertebrate immunity. But all in all this type of review is of enough interest to warrant inclusion in a journal. One minor thing is that the activation of the proPO cascade and that of the Toll pathway shares the same proteinases as was first shown in Tenebrio by Bok Luel Lee in 2008-2009 and later basically confirmed in both Manduca sexta and Drosophila.
Response: we added this information and relevant references
Shan T, Wang Y, Bhattarai K, Jiang H. An evolutionarily conserved serine protease network mediates melanization and Toll activation in Drosophila. Sci Adv. 2023 Dec 22;9(51):eadk2756. doi: 10.1126/sciadv.adk2756.
Wang Y, Yang F, Cao X, Huang R, Paskewitz S, Hartson SD, Kanost MR, Jiang H. Inhibition of immune pathway-initiating hemolymph protease-14 by Manduca sexta serpin-12, a conserved mechanism for the regulation of melanization and Toll activation in insects. Insect Biochem Mol Biol. 2020 Jan;116:103261. doi: 10.1016/j.ibmb.2019.103261.
Kan H, Kim CH, Kwon HM, Park JW, Roh KB, Lee H, Park BJ, Zhang R, Zhang J, Söderhäll K, Ha NC, Lee BL. Molecular control of phenoloxidase-induced melanin synthesis in an insect. J Biol Chem. 2008 Sep 12;283(37):25316-25323. doi: 10.1074/jbc.M804364200.
Reviewer 3 Report
Comments and Suggestions for Authors
Despite being evolutionarily older and serving as a precursor to vertebrate immunity, invertebrate immunity is far from simplistic. While lacking lymphocytes and functional immunoglobulins, the invertebrate immune system possesses a multitude of sophisticated mechanisms and features that were traditionally believed to be exclusive to adaptive immunity. One such example is the presence of long-term immune memory, which has long been associated solely with adaptive immune responses. This revelation challenges the prevailing understanding of immune memory and calls for a closer examination of the complexities within the invertebrate immune system.
In this comprehensive review, authors aim to explore the cellular and molecular aspects of invertebrate immunity, shedding light on various intriguing phenomena. These include the epigenetic foundations of innate memory, the transgenerational inheritance of immunity, the genetic mechanisms that confer immunity against invading transposons, the intricate mechanisms of self-recognition, the fascinating dynamics of natural transplantation, and the intriguing interplay of germ/somatic cell parasitism. By delving into these captivating aspects, authors want to deepen our understanding of the remarkable immune system of invertebrates and its broader implications for the field of immunology.
In the field of immunology, the study of the immune system has predominantly focused on vertebrates, which possess both innate and adaptive branches of immunity. However, invertebrates, with their exclusive reliance on innate immunity, offer a unique opportunity to investigate the cellular and molecular mechanisms of innate immunity without the confounding influence of adaptive immunity. This distinguishing characteristic makes them an ideal model system for understanding the intricate workings of innate immunity in its purest form.
I have almost no criticisms of the structure and content of your review article, although I have noticed a few minor oversights:
- Line 43 - "recent genomic studies" - It would be beneficial to include references to these studies.
- Lines 77-78 - "(Buchmann, 2014)" - Please incorporate this citation into the overall citation style and provide it in the reference list.
- Lines 78-79 - Since these shrimp species have not been mentioned earlier in the text, please provide their full generic names: "Penaeus monodon, Litopenaeus vannamei, Marsupenaeus japonicus"
- Line 148 - "Insects, spiders, arthropods, and mollusks" - It is not entirely accurate as arthropods include insects and spiders. Please rephrase accordingly.
- Lines 492-493 - It is necessary to provide the definition of PIWI and piRNAs upon their first mention (lines 336-337).
- Line 496 - What do you mean by "piRNA pathway"? In the classical understanding, it is not entirely accurate to state that "piRNA pathway operates in all organisms." For example, plants do not have PIWI proteins.
- If possible, please provide Figures 1 and 2 in a higher resolution in the final version.
- While not critical, for better readability, it is recommended to avoid hyphenation, especially in titles.
Best regards,
L.
Author Response
Reviewer 3
I have almost no criticisms of the structure and content of your review article, although I have noticed a few minor oversights:
- Line 43 - "recent genomic studies" - It would be beneficial to include references to these studies.
Response: as requested we added the references
Loker ES, Adema CM, Zhang SM, Kepler TB. Invertebrate immune systems--not homogeneous, not simple, not well understood. Immunol Rev. 2004 Apr;198:10-24. doi: 10.1111/j.0105-2896.2004.0117.x.
- Lines 77-78 - "(Buchmann, 2014)" - Please incorporate this citation into the overall citation style and provide it in the reference list.
Response: we changed this citation to the number, which was already in the reference list
- Lines 78-79 - Since these shrimp species have not been mentioned earlier in the text, please provide their full generic names: "Penaeus monodon, Litopenaeus vannamei, Marsupenaeus japonicus"
Response: As requested, we added the full names
- Line 148 - "Insects, spiders, arthropods, and mollusks" - It is not entirely accurate as arthropods include insects and spiders. Please rephrase accordingly.
Response: Thanks for noticing this error, we corrected the statement
- Lines 492-493 - It is necessary to provide the definition of PIWI and piRNAs upon their first mention (lines 336-337).
Response: as requested, we provided these definitions
- Line 496 - What do you mean by "piRNA pathway"? In the classical understanding, it is not entirely accurate to state that "piRNA pathway operates in all organisms." For example, plants do not have PIWI proteins.
Response: we changed this statement
- If possible, please provide Figures 1 and 2 in a higher resolution in the final version.
Response: The original figures are in a high resolution
- While not critical, for better readability, it is recommended to avoid hyphenation, especially in titles.
Response: thank you, noted
